# Optimization and Design of Disc-Type Furrow Opener of No-Till Seeder for Green Manure Crops in South Xinjiang Orchards

**Rui Ye** [1,2], **Xueting Ma** [1,2], **Jinfei Zhao** [1,2], **Jiean Liao** [1,2,*], **Xinying Liu** [1,2], **Linqiao Xi** [3] and **Guangdong Su** [1,2]

1    College of Mechanical Electrification Engineering, Tarim University, Alar 843300, China; yrhoode@163.com (R.Y.); 120160004@taru.edu.cn (X.M.)
2    Agricultural Engineering Key Laboratory, Ministry of Higher Education of Xinjiang Uygur Autonomous Region, Tarim University, Alar 843300, China
3    College of Animal Science, Tarim University, Alar 843300, China
*    Correspondence: 120100010@taru.edu.cn; Tel.: +86-158-8684-1796

**Abstract:** For the issues of the poor stability of the furrow opener depth, large soil backfill depth, and inconsistent furrow shape on a no-till seeder for planting green manure between rows of orchards in South Xinjiang, a double-disc, corrugated furrow opener is designed. This paper analyzes the law of soil movement between corrugated double-disc and traditional double-disc furrow openers using the discrete element method (DEM) and concludes that the corrugation width and number of corrugations on the corrugated double-disc furrow opener are the primary factors affecting furrowing operation. When the number of corrugations is sixteen, the forward speed is six kilometers per hour, and when the corrugation width is seventeen and a half millimeters, the simulation operation parameters are optimal. The soil-bin validation experiment results are as follows: Under the condition of an 80 mm furrow depth, the stability of the average furrow depth is enhanced by 3.54%, the working resistance and the average disrupted soil area are increased by 26.16 N and 220 mm$^2$, respectively, and the backfill depth is decreased by 10.98 mm. The operation effect of a double-disc furrow opener with corrugated discs is enhanced by the high stability of the furrow depth, low working resistance, and small backfill depth. This study provides a theoretical foundation for the design and optimization of the furrow opener components of a no-till seeder for planting green manure between rows of orchards.

**Keywords:** discrete elements; simulation tests; no-till furrowing; soil-bin tests





## 1. Introduction

Xinjiang, located along China's northwest frontier, has a typical continental climate [1]. In Xinjiang, the large temperature difference between day and night has improved the quality of forests and fruits, whereas the lack of water resources and low soil organic matter severely impedes the development of forest and fruit industries [2]. Green manure's cultivation area is expansive, reaching 10 million to 15 million hm$^2$ at its zenith, and its development space is approximately 46 million hm$^2$. The widespread practice of planting green manure plants in orchards for soil management purposes green manure can enhance the organic matter and structure of soil, as well as the trace element content of soil, the growth and development of fruit trees, and fruit quality [3]. In the no-tillage sowing of green manure, the furrow opener is one of the key components of a no-tillage seeder, and its performance impacts the quality of no-tillage sowing, which is the crucial link in green manure planting. It is necessary to further enhance no-till furrowing components in South Xinjiang orchards due to the unique sowing method and soil conditions.

Existing furrow components for green manure sowing are primarily double-disc furrow openers, which are extensively utilized in no-till seeders due to their low working

resistance and high passability. Due to their high quality and insufficient furrow depth stability, double-disc furrow openers are unsuitable for small and medium-sized no-till seeders. It is essential for agricultural production to continually optimize the structure of double-disc furrow openers to make them more in line with agronomic requirements, improve the stability of sowing depth, and reduce the backfill depth and soil disturbance under the condition of no-tillage sowing of green manure while minimizing working resistance and backfill.

DEM can be used to simulate particles and study the microscopic and macroscopic changes between materials, and it is widely used in the structural design of furrow openers [4]. Scholars usually take working resistance and soil disturbance as the evaluation index of no-tillage sowing furrow opener performance and have made a lot of achievements in their research [5]. Wang Yueming et al. [6] designed a bionically coupled disc furrow opener, which was inspired by digging the convex hull of animals such as dung beetle chests and the scales of pangolin back ridges as bionic coupling elements, which reduced the resistance and energy consumption of the disc furrow opener. The influence of different structures on forward and lateral soil disturbance was analyzed by DEM. The results showed that under the same experimental conditions, the furrow resistance of the bionic coupled furrow opener was obviously lower than that of the common flat furrow opener, and it can be used for high-efficiency agricultural farming. Ahmad Fiaz et al. [7] made discrete element simulations on a notch, tooth-shaped, and double-disc furrow opener by using EDEM and studied the performance of a disc furrow opener in paddy soil and field experiments. A 3D EDEM model was established, and the Hertz–Mindlin contact model and bonding effect were used to simulate the bonding obligation between soil moisture and cohesive particles. Through a comparison of simulation and field experiment data, the applicability of each furrow opener under different working conditions was determined. Zhou Wenqi et al. [8] designed a bionic furrow opener for the deep application of liquid fertilizer according to the characteristics of low power consumption of a badger's claw toe structure to improve the operation performance of the deep application of liquid fertilizer. This simulation experiment was carried out by using DEM, and a laboratory soil-bin test was carried out. It was concluded that the operation angle, operation speed, and fertilization depth of the bionic liquid fertilizer deep application furrow opener had a significant influence on the reduction in power and energy consumption. The operation performance of the bionic liquid fertilizer deep application furrow opener was obviously better than other types of liquid fertilizer deep application furrow openers. Saeys et al. [9] explored two cutting methods for a furrow opener—sliding cutting and chopping cutting—and reached the conclusion that the working resistance is small when sliding cutting is used.

EDEM was used to optimize the furrow components of a no-till seeder in the aforementioned study. On the basis of the discrete element simulation model of a soil furrow opener, a discrete element simulation test of various furrow opener varieties was conducted. Using the law of falling soil movement, the primary factors affecting the stability of the furrow depth and backfill depth were analyzed in order to enhance the stability of the furrowing depth and reduce working resistance and backfill depth. A discrete element simulation test confirmed the rationality of the proposed design. The experiments were conducted in a soil bin, and the operation effects were contrasted, providing a theoretical basis for the design and optimization of the furrow sections of a green manure no-tillage seeder.

## 2. Materials and Methods

### 2.1. Determination of the Structural Parameters of the Corrugated Double Disc

2.1.1. Determination of the Diameter of the Disc

As shown in Figure 1, the integral structure of the corrugated double-disc opener was identical to that of the traditional double-disc opener, but the structural parameters diverged due to the different disc types. The interaction between the corrugated disc and the rapeseed stalk was analyzed and clarified.

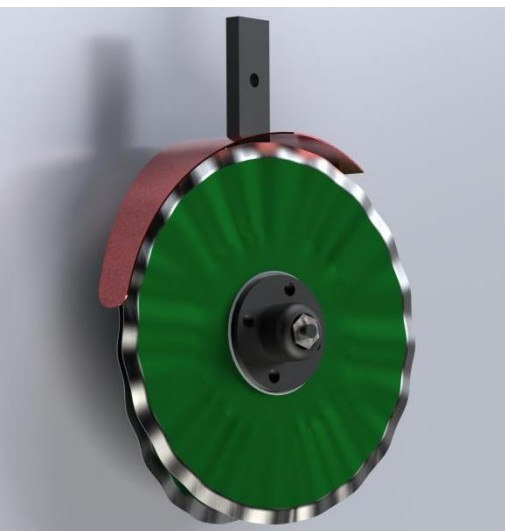

**Figure 1.** Corrugated double-disc furrow opener.

Using rapeseed stalk as the research object, the force situation at the moment of disc contact was examined, as shown in Figure 1.

As shown in Figure 2, $O_P$ is the center of the disc; $O_J$ is the center of the rapeseed stalk; $v$ is the horizontal motion speed of the furrow opener, m/s; $\omega_P$ is the angular velocity of the disc, rad/s; $\alpha$ is the pressure angle (°); $F_T$ is the force of the disk on the rapeseed stalk, N; $F_N$ is the support force of the ground on the rapeseed stalk, N; $G_1$ is the gravity of the rapeseed stalk, N; $f_1$ is the disk frictional force on the rapeseed stalk, N; $f_2$ is the frictional force on the rapeseed stalk from the ground, N.

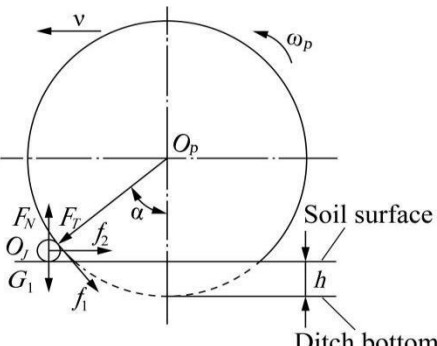

**Figure 2.** Force analysis of rapeseed stalk.

From the static equilibrium analysis of rapeseed stalk, it follows that

$$\begin{cases} f_1 \sin\alpha + F_t \cos\alpha + G_1 = F_N \\ f_1 \cos\alpha + f_2 = F_T \sin\alpha \end{cases} \tag{1}$$

It can be obtained from the geometric relationship that

$$\cos\alpha = 1 - \frac{2h + d}{D} \tag{2}$$

where $D$ is the diameter of the disc, mm; $d$ is the diameter of the rapeseed stalk, mm.

If the rapeseed stalks are cut by the disc and do not migrate and clog by pushing, the following needs to be met:

$$f_1 \cos\alpha + f_2 \geq F_T \sin\alpha \tag{3}$$

By introducing the friction angle, $\varphi_1$, between the rapeseed stalk and the disc and the friction angle, $\varphi_2$, between the rapeseed stalk and the ground surface, the joint (1), Equations (2) and (3) can be obtained.

$$\cos^{-1}(1 - \frac{2h + d}{D}) \le \phi_1 + \phi_2 \tag{4}$$

Considering the green manure planting conditions in southern Xinjiang orchards, the average diameter of rapeseed stalks is 8 mm [10], and combined with test measurements, in this design, the diameter of the stalks is assumed to be $d = 8$ mm. The depth in the soil when the disc is opened should not be too large; otherwise, it will not be able to cut the surface stalks and cause the machine to block [11], so we assumed an opening depth $h = 80$ mm. The friction angle is 16° to 21° [12]; taking into account the limit situation, a maximum value of $\varphi_1 = 21°$ is considered in the calculation. Considering the fact that the device is mainly used for no-till sowing and furrowing operations in arid areas where the surface soil moisture (about 13%) is low, the average friction angle, $\varphi_2$, between the rapeseed stalk and the surface was determined to be 19°. A combination of the above parameters, from Formula (4), can be obtained from a disc diameter of $D \ge 230$ mm.

An instantaneous velocity analysis of a disc cutting a rapeseed stalk during a furrowing operation is shown in Figure 2. The analysis approximates the disc motion as pure rolling, with point $A$ as the instantaneous cutting circle center and $r_{AB}$ as the instantaneous cutting radius, to establish a mathematical model of rapeseed stalk cutting [13].

In Figure 3, $A$ is the instantaneous cutting circle center; $B$ is the instantaneous contact point between the disc and the stalk; $C$ is the projection point of point $B$ on the path $O_PA$; $\alpha$ is the pressure angle (°); $\omega_{AB}$ is the instantaneous cutting angular velocity, rad/s, with point $A$ as the circle center; $v_B$ is the instantaneous cutting linear velocity, m/s, with point $A$ as the circle center.

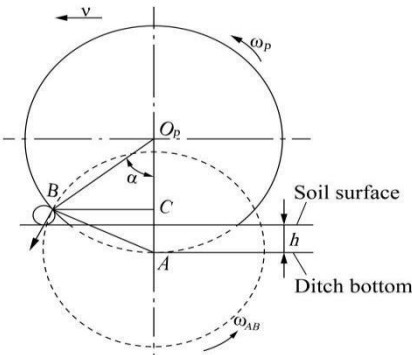

**Figure 3.** Analysis of the instantaneous speed of the disc cutting the stalk.

Analysis of the instantaneous velocity relationship between the discs cutting the stalk gives the following:

$$\begin{cases} v_B = \omega_{AB} r_{AB} \\ v = \omega_p \frac{D}{2} \end{cases} \tag{5}$$

where $r_{AB}$ is the instantaneous cutting radius, mm.

The analysis of the geometric relationship gives the following:

$$r_{AB}^2 = r_{OpA}^2 + r_{OpB}^2 - 2r_{OpA}r_{OpB} \cos \alpha \tag{6}$$

The instantaneous angular velocity of the cut at point $B$ and the angular velocity of the disc are equal in magnitude, i.e.,

$$\omega_{AB} = \omega_p \tag{7}$$

Coupling (2), (5), (6), and (7) gives the following:

$$v_B = 2\omega_P \sqrt{2(2h + d)} \cdot \sqrt{D} \tag{8}$$

Bringing the furrowing depth $h$ = 80 mm and the stalk diameter $d$ = 8 mm into Equation (8) gives the following:

$$v_B = 2\sqrt{390} \cdot \omega_P \sqrt{D} \tag{9}$$

As can be seen from Equation (9), under the condition that the angular speed of the disc, $\omega_P$, is certain, the instantaneous cutting line speed, $v_B$, of the disc cutting rapeseed stalks is positively related to the diameter of the disc, $D$, i.e., the larger the diameter of the disc, the better the cutting effect of the disc on the stalks and the less the seed racking and seed drying phenomenon after the disc furrowing. As can be seen from Equation (2), when the furrowing depth and stalk diameter are definite values, the larger the disc diameter, $D$, the smaller the pressure angle, $\alpha$. Increasing the disc diameter, $D$, is conducive to cutting off the rapeseed stalks covered by the ground; however, if the disc diameter is too large, this will limit the structure of the whole machine and the height of the frame, as the main function of the corrugated double-disc opener in the furrowing process is to bring the uncut stalks to the sides through the corrugation, with the design usually taken as 200–300 mm [14].

The corrugated discs cut the stalks and stubble in the seed belt and do not need to have a strong cutting capacity. Combined with the above analysis, the corrugated disc diameter was taken to be a median value of 250 mm in the design, i.e., a corrugated disc diameter of $D$ = 250 mm [15].

### 2.1.2. Determination of the Location of the Gathering Point and the Width of Furrowing

In a double-disc furrow opener, the edges of the double discs intersect at a point below the front, which is called the gathering point of the double discs, $m_0$, the position of the gathering point is indicated by $\beta_0$ and the double discs form an angle, $\varphi_0$, the structural relationship of which is shown in Figure 4. The choice of the point of aggregation is determined by the radius of the double disc and the agronomic requirements of the sowing depth.

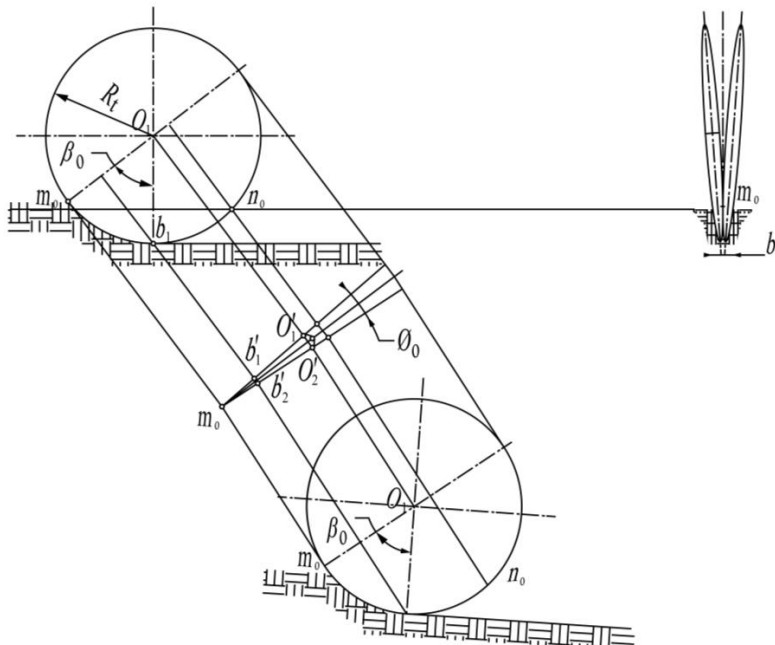

**Figure 4.** Position and angle of the double-disc convergence point.

The radius of the discs and the angle of the double discs of the double-disc opener are important factors affecting the depth of furrowing, and the formula for calculating the width of the furrowing is as follows:

$$K = R_t(1 - \cos \beta) \sin \frac{\phi}{2} \tag{10}$$

where $R_t$ is the diameter of the disc, mm; $\varphi$ is the angle between the two discs, °; $\beta$ is the angle between the aggregation point, m, and the vertical direction, °.

As can be seen from the above equation, when the disc radius and double-disc nip angle increase, the width of the furrow opener opening increases and so does the bulge in the seed furrow, seriously affecting sowing consistency. Therefore, the position of the double-disc gathering point and the double-disc clamping angle should be chosen according to the size of the seed [16].

According to the literature and analyses, the radius of the double-disc opener is too large for the installation size, the bulge in the seed furrow increases, the bottom of the seed furrow is poorly flattened and affects the sowing effect; if the radius of the disc is too small, the double-disc opener is prone to inflexible rotation and congestion when working, which increases the working resistance and leads to a decrease in work quality. Therefore, the radius of the corrugated double-disc opener should meet the following:

$$\begin{cases} R_t > h \\ \frac{R_t - h}{R_t} = \cos \beta_0 \end{cases} \tag{11}$$

where $h$ is the depth of furrowing, mm.

Due to the agronomic requirements of strip sowing for green manure planting in orchards, the sowing depth is 5 to 18 mm. According to the results of field research, farmers mostly choose a planting depth of about 80 mm. Considering the installation space of the furrow opener and seed rower and the depth of the furrow, the disc radius of the double-disc opener, $R_t$ = 125 mm, can be determined and the angle of the gathering point is 53°.

As can be seen from Equation (10), the opening width, $b$, is related to the radius of the disc, $R_t$, and the angle, $\varphi_0$, of the double discs. In order to ensure the opening passability of the corrugated double-disc opener and the working performance of the device, the opening width of the corrugated double-disc opener should be made as small as possible [17]. In combination with the installation structure of the seed rower and the furrow opener, the opening of the double disc should be sufficient to fit the drive parts of the seed rower so that the green manure seeds can fall smoothly into the opened furrow and the furrow width should meet the following:

$$\begin{cases} b < W_{\max} \\ 2R_t \sin \frac{\varphi_0}{2} > 19 \end{cases} \tag{12}$$

where $W_{\max}$ is the maximum value of green manure seed width.

Taking the triaxial size of a typical green manure oat seed as an example, it can be found that $W_{\max}$ = 5.80 mm. Coupling (10)~(12), it can be obtained that the double-disc clamping angle, $\varphi_0$ > 8.32°; thus, it can be determined that the double-disc clamping angle, $\varphi_0$ = 9°.

### 2.2. Discrete Element Soil Particle Movement Analysis

2.2.1. Analysis of Soil Particle Movement Law

To ensure the operating range of the working parts, a soil simulation model area of 2200 mm length, 300 mm width, and 150 mm height was established in this study. The soil particle models were set as single-particle spherical, double-particle columns, three-particle models, and agglomerate models formed by stacking four particles [18]. According to the reviewed literature, other soil model parameters are shown in Table 1, with the soil particle generation time ranging from 0 to 3 s and the settlement time being 1 s [19].

**Table 1.** Soil model parameters.

| Parameters | Numerical Values |
| --- | --- |
| Poisson's ratio | 0.38 |
| Density (kg·m$^{-3}$) | 1850 |
| Shear modulus (MPa) | $1.24 \times 10^6$ |
| Soil interparticle recovery factor | 0.2 |
| Coefficient of static friction between soil particles | 0.4 |
| Rolling friction coefficient between soil particles | 0.3 |
| Bonding radius (mm) | 9.24 |
| Critical tangential stress (Pa) | $6.8 \times 10^4$ |
| Critical normal stress (Pa) | $2.0 \times 10^5$ |
| Normal contact bond stiffness between soil particles (N·m$^{-1}$) | $3.4 \times 10^8$ |
| Tangential contact bond stiffness between soil particles (N·m$^{-1}$) | $1.5 \times 10^8$ |

According to the agronomic requirements of no-till sowing, the no-till sowing procedure comprises residue breaking, seed furrowing, seed release, mulching, and suppression. In order to thoroughly examine the soil movement law during the furrow opener procedure, the double-disc furrow opener and the corrugated double-disc furrow opener were chosen for this investigation. The simulation geometry model of the furrow opener device was developed with reference to the Agricultural Machinery Design Manual, and the two geometry models were saved in .x_t format and imported into the soil green manure stalk discrete element simulation model. The material of the furrow opener device was 65 Mn steel, and based on the reviewed literature, the steel density was set at 7865 kg/m$^3$, Poisson's ratio at 0.3, shear modulus at $7.97 \times 10^{11}$ Pa, and other contact parameters as shown in Table 2. Additionally, the furrow depth into the soil was set at 80 mm, forward speed of 4 km/h, and rotation speed of 5 rad/s [20].

**Table 2.** Selected contact parameters of the discrete element model.

| Parameters | Numerical Values |
| --- | --- |
| Steel—soil static friction coefficient | 0.65 |
| Steel—coefficient of dynamic soil friction | 0.11 |
| Steel—soil recovery factor | 0.60 |

A selection of any six soil particles within the working depth in the vertical direction in the front of the furrow opener was made. The particle movement and soil particle movement were separately output. As shown in Figure 5a, the front section of the corrugated double-disc furrow opener first encounters the soil particles, resulting in an oblique downward movement trace along the advancing direction. After the furrow opener has passed, as shown in Figure 5b, the soil particles are elevated by the action, resulting in an oblique upward movement traced along the advancing direction. As shown in Figure 5c, the drag causes the soil particles to advance with the furrow opener, but their movement speed gradually decreases due to the adjacent soil's resistance. As shown in Figure 5d, as the furrow opener advances, the soil particles progressively retreat until a stable position is reached. As shown in Figure 5e, the total movement trace of soil particles under the influence of the furrow opener is press–lift–begin to fall back–fall back–fall back to stability.

2.2.2. Analysis of Soil Particle Movement Patterns

The soil particle simulation time was 0.5 s from the beginning of the furrow opener (6.0 s) to its end (6.5 s), where 6.0–6.2 s represents the effect of the furrow opener on the particles and 6.2–6.5 s represents the effect of the selected particles on the surrounding particles after the furrow opener has passed through. According to Figures 6–9, after the transit of the corrugated double-disc furrow opener, the external force of soil particles decreased to 6.2–6.5 s. The soil particles were predominantly compressed by the surrounding soil,

and the value of the force hardly altered, while the soil's kinetic energy and moving speed decreased and then increased. Nevertheless, after passing through the traditional double-disc furrow opener, the force value changed significantly, the kinetic energy remained unchanged, and the moving speed decreased. As shown in Figure 10, the amount of change in velocity and kinetic energy of soil particles moving under the action of corrugated discs during the simulated operating time period was much higher than for soil particles under the action of traditional double discs. It can be concluded that the disc corrugation can effectively hurl the soil to both sides of the seed furrow, and the soil can retain its kinetic energy and movement speed after the furrow opener has passed (Figure 10a). In contrast, the traditional double-disc furrow opener only compresses the soil particles, and the furrow opener backfills the seed furrow under the influence of surrounding particles (Figure 10b), so disc corrugation can effectively reduce the furrowing resistance of the furrow opener and the amount of backfill from the seed furrow.

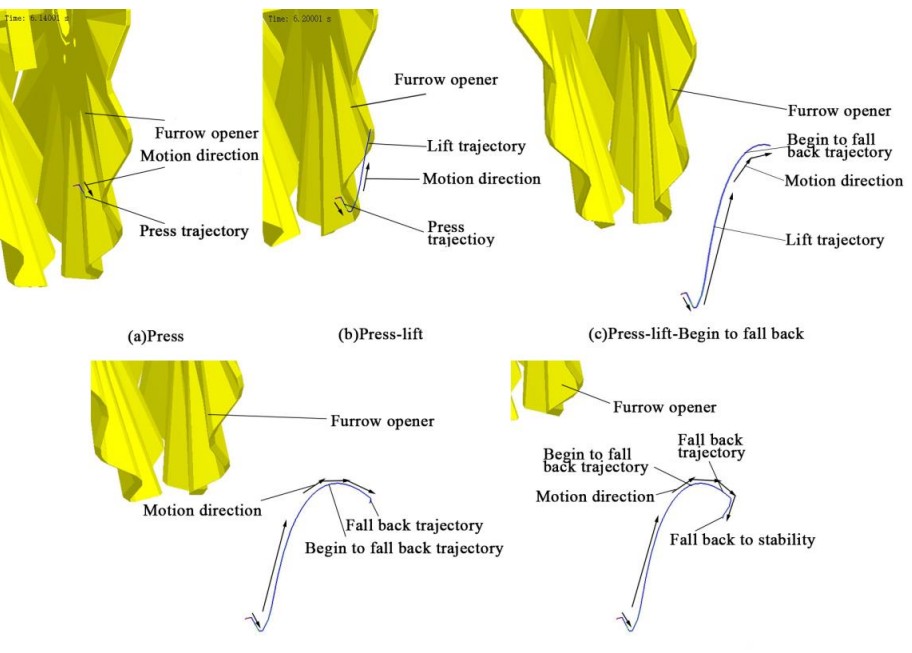

**Figure 5.** Soil simulation trajectory.

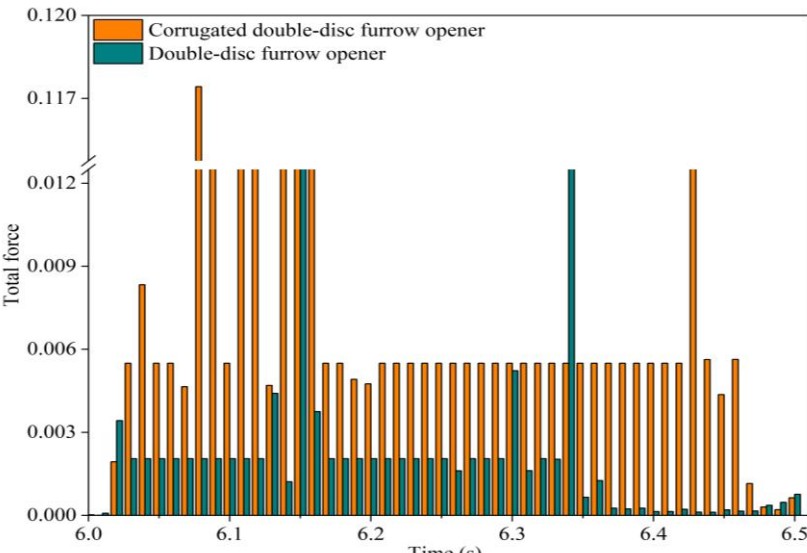

**Figure 6.** Time comparison of changes under combined external forces on soil particles.

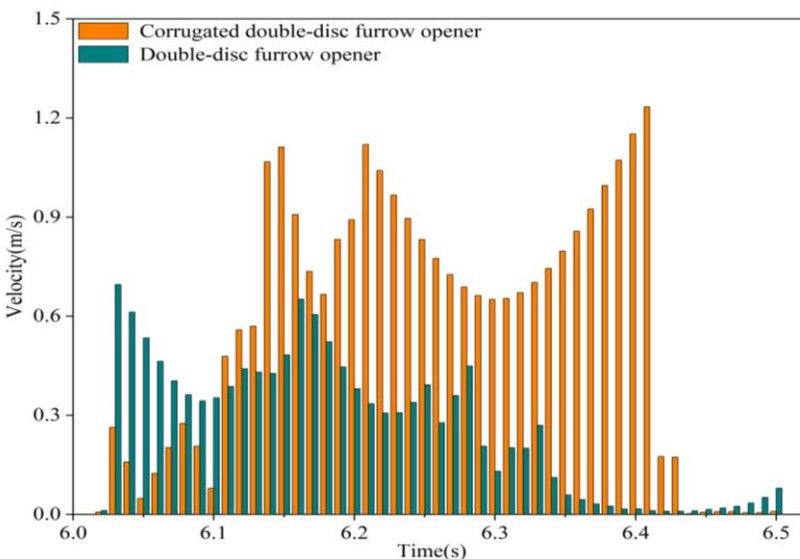

**Figure 7.** Time comparison of changes in soil particle movement velocity.

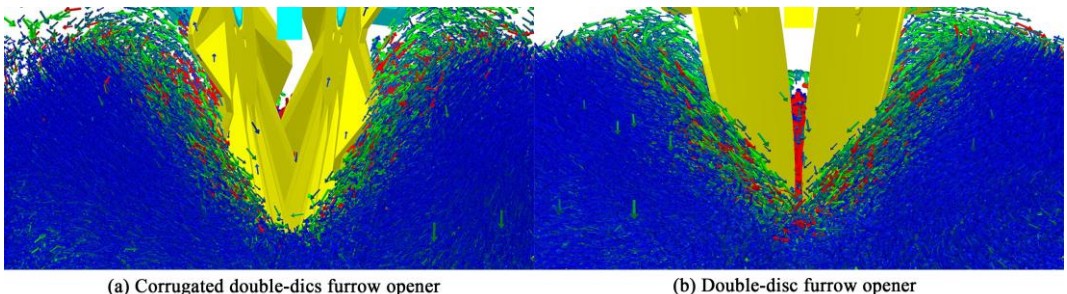

**Figure 8.** Illustration of the direction of force on soil particles under the action of different openers.

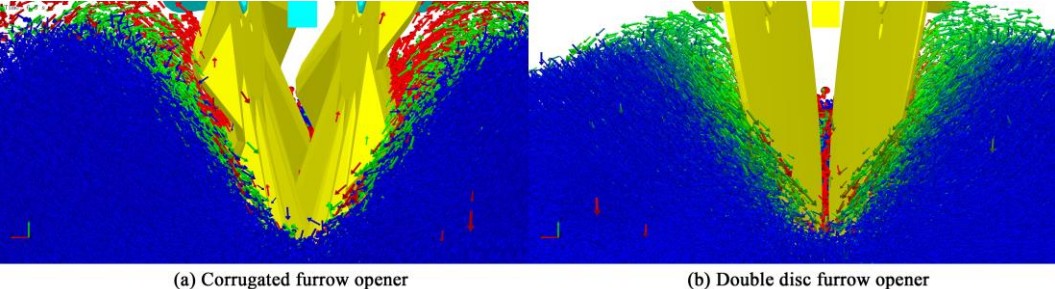

**Figure 9.** Schematic diagram of the direction of velocity of soil particles under the action of different furrow openers.

### 2.3. Simulation Test Method

From the analysis of the trajectory and movement law of soil particles, it can be deduced that the number of corrugations and the corrugation width of the corrugated discs are the most influential factors for enhancing the stability of the furrow depth and minimizing the working resistance and soil backfill depth of the furrow opener. As the evaluation criteria, the stability of furrow depth, $y_1$; backfill depth, $y_2$; and working resistance, $y_3$, were utilized. The simulation stroke was 2.2 m, the forward speed was 4 km/h, and the direction was along the negative *y*-axis. Each group of experiments was conducted five times, and the average of the results was calculated. The levels of the test factor are shown in Table 3.

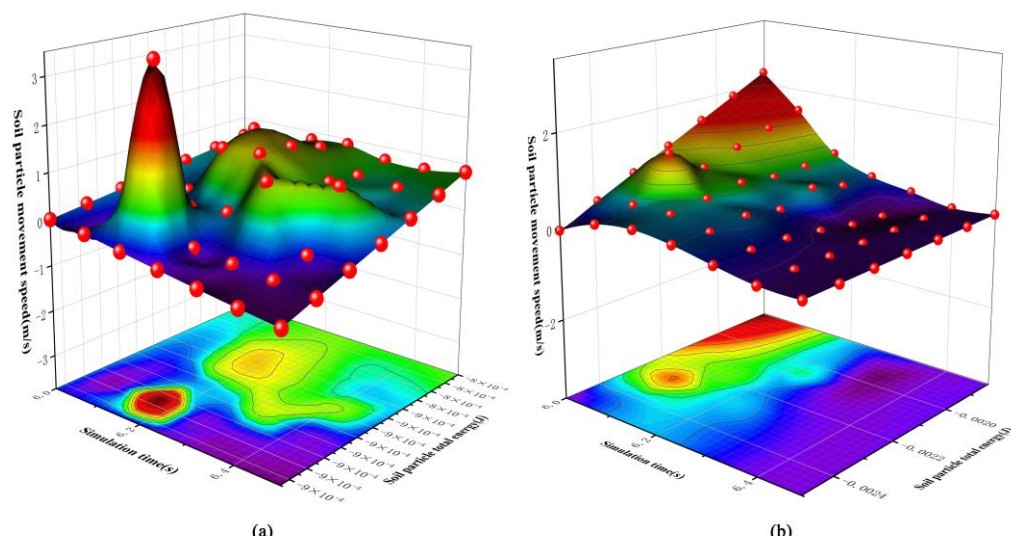

**Figure 10.** Effect of furrow opener on the velocity of soil particle movement and total kinetic energy of soil particles. (**a**) Corrugated double disc. (**b**) Double-disc furrow opener.

**Table 3.** Test factor coding table.

| Code Value | Factors | | |
|---|---|---|---|
| | Number of Corrugations L1 | Forward Speed L2 (km/h) | Corrugation Width L3 (mm) |
| 1.682 | 17 | 9 | 19 |
| 1 | 16 | 8 | 17.5 |
| 0 | 14 | 6 | 15 |
| −1 | 12 | 4 | 12.5 |
| −1.682 | 11 | 3 | 11 |

*2.4. Soil-Bin Validation Experiment*

2.4.1. Test Conditions and Apparatus

A soil-bin comparison experiment was conducted in September 2022 at the modern agricultural machinery experimental training base of Tarim University in Alar, Xinjiang. The selected test area was 15 m × 4 m. The soil bulk weight (0–100 mm soil layer) in the test area was 1.62 g/cm$^3$, and the soil moisture content (0–150 mm) was (13 ± 1)%.

The main test materials for the soil-bin experiment were as follows: soil-bin test bench, furrow opener, tape measure, tension sensor, sensor data acquisition card, corrugated double-disc furrow opener, traditional double-disc furrow opener, etc. As shown in Figure 11.

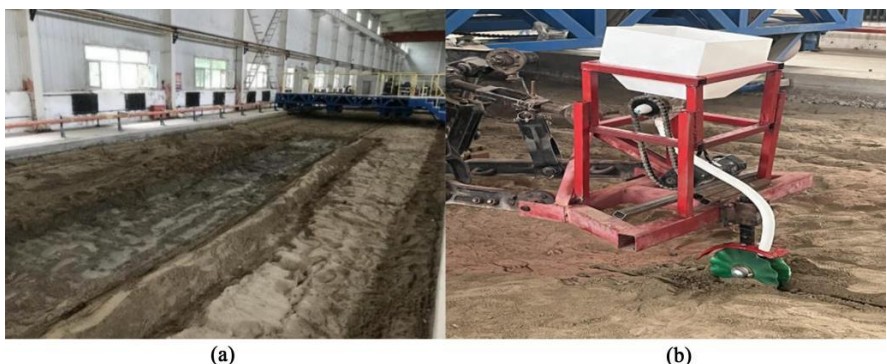

**Figure 11.** Soil trough comparative validation test environment. (**a**) Test area. (**b**) Schematic diagram of the stand.

2.4.2. Experiment Method

The main objective of the comparative validation experiment was to compare the performance of the corrugated double-disc furrow opener with that of the traditional double-disc furrow opener under specific test conditions. Furrow depths of 60, 80, and 100 mm were set for the two types of openers, with a forward speed of 6 km/h. The stroke of each set of tests was set at 15 m, and each set of tests was repeated five times.

The size of the soil disturbance area was expressed as the cross-sectional area of soil disturbed on both sides of the furrow after the furrow opener operation. The center bubble of the horizontal ruler was fixed to the furrow profile meter, the shape of the seed furrow was outlined on the coordinate paper, and the soil disturbance area was calculated according to the shape of the seed furrow on the coordinate paper [21]. The soil furrow profile is shown in Figure 12.

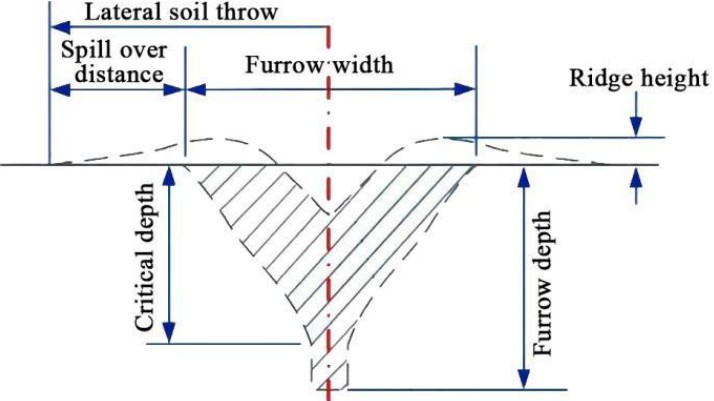

**Figure 12.** Contours of the soil furrow pattern.

**3. Results and Analysis**

*3.1. Analysis of Simulation Results*

Design-Expert 8.0.6 software was used to fit the test data shown in Table 4, with multiple regressions and ANOVAs performed to remove non-significant terms, which resulted in regression equations for the effects of each factor on the furrowing depth stability coefficient, $y_1$; the depth of soil backfill, $y_2$; and the working resistance, $y_3$ [22].

$$y_1 = 89.89 - 1.71X_1 - 1.98X_2 + 1.46X_3 + 1.09X_1X_2 \\ +0.73X_1X_3 + 0.93X_2X_3 - 1.64X_1^2 - 1.89X_2^2 - 0.29X_3^2 \tag{13}$$

$$y_2 = 32.67 - 4.63X_1 - 1.98X_2 + 2.48X_3 + 1.13X_1X_2 \\ -1.62X_1X_3 - 2.12X_2X_3 - 1.34X_1^2 + 1.85X_2^2 - 0.98X_3^2 \tag{14}$$

$$y_3 = 520.83 + 76.66X_1 + 49.14X_2 + 47.35X_3 + 32.72X_1X_2 \\ -46.20X_1X_3 - 7.75X_2X_3 - 60.79X_1^2 + 95.05X_2^2 - 9.98X_3^2 \tag{15}$$

For a more intuitive analysis of the relationship between the evaluation index and the test factors, the response surface was generated with Design-Expert 10 software. According to the regression equation and response surface presented above, an interaction exists between the number of corrugations, forward speed, and corrugation width [23].

As shown in Figure 13a, when the current feeding speed remains constant, the furrowing depth stability, $y_1$, first increases and then decreases as the number of corrugations increases. This is because the corrugated disc fails to effectively remove the seed furrow soil when the number of corrugations is low. The soil disturbance coefficient in the seed furrow is low because the corrugated disc effectively loosens the seed furrow soil as the number of corrugations increases, and the furrow depth becomes more stable. When the number of corrugations is excessive, the disturbance domain of the corrugated disc is large,

and the quantity of backfill is large and irregular, which reduces the furrowing depth's stability. When the number of corrugations remains constant, the stability of the furrow depth increases prior to decreasing as advancing speed increases. The reason for this is that when the advancing speed is low, the furrow type is stable following the operation of the corrugated disc. As the advancing pace increases, the quantity of backfill decreases, and the depth stability of the furrow increases. However, when the present advancing pace is excessive, the bottom of the furrow is raised and the furrow type varies, resulting in a decrease in the depth stability. In the interaction between the number of corrugations and forward motion, the number of corrugations has the greatest effect on the stability of the furrow depth.

**Table 4.** Simulation experimental results.

| Test Serial Number | Factors | | | Indicators | | |
| --- | --- | --- | --- | --- | --- | --- |
| | Number of Corrugations L1 | Forward Speed L2 (km/h) | Corrugation Width L3 (mm) | Depth Stability of Furrowing (%), $y_1$ | Depth of Soil Backfill (mm), $y_2$ | Working Resistance (N), $y_3$ |
| 1 | −1 | −1 | −1 | 89.44 | 34 | 469.70 |
| 2 | 1 | −1 | −1 | 84.15 | 26 | 683.87 |
| 3 | −1 | 1 | −1 | 81.96 | 31 | 531.43 |
| 4 | 1 | 1 | −1 | 80.43 | 29 | 819.83 |
| 5 | −1 | −1 | 1 | 89.84 | 45 | 726.96 |
| 6 | 1 | −1 | 1 | 86.86 | 32 | 699.68 |
| 7 | −1 | 1 | 1 | 85.48 | 35 | 701.03 |
| 8 | 1 | 1 | 1 | 87.45 | 25 | 861.29 |
| 9 | −1.682 | 0 | 0 | 90.36 | 38 | 542.12 |
| 10 | 1.682 | 0 | 0 | 81.14 | 20 | 786.75 |
| 11 | 0 | −1.682 | 0 | 88.64 | 41 | 660.92 |
| 12 | 0 | −1.682 | 0 | 81.45 | 35 | 861.72 |
| 13 | 0 | 0 | −1.682 | 87.69 | 25 | 415.93 |
| 14 | 0 | 0 | 1.682 | 91.46 | 35 | 512.60 |
| 15 | 0 | 0 | 0 | 89.24 | 32 | 491.78 |
| 16 | 0 | 0 | 0 | 90.17 | 33 | 486.99 |
| 17 | 0 | 0 | 0 | 90.56 | 31 | 495.94 |
| 18 | 0 | 0 | 0 | 89.89 | 34 | 549.90 |
| 19 | 0 | 0 | 0 | 90.03 | 35 | 563.67 |
| 20 | 0 | 0 | 0 | 89.25 | 31 | 546.40 |
| 21 | 0 | 0 | 0 | 91.07 | 30 | 488.99 |
| 22 | 0 | 0 | 0 | 86.98 | 31 | 520.22 |
| 23 | 0 | 0 | 0 | 89.56 | 32 | 488.16 |

As shown in Figure 13b, when the corrugation width is held constant, the stability of furrow depth, $y_1$, initially increases and then decreases with the number of corrugations; when the number of corrugations is held constant, the stability of the furrow depth, $y_1$, increases with the corrugation width. The reason for this is that as the width of the corrugation increases, the amount of soil pushed out by the corrugated disc increases and the amount of backfill decreases, thereby enhancing the stability of the furrowing depth. In the interaction between the number of corrugations and the width of corrugations, the width of corrugations has the greatest effect on the stability of the furrowing depth, $y_1$.

As shown in Figure 13c, when the number of corrugations remains constant, the backfill depth, $y_2$, increases slowly with the increase in corrugation width because the disturbance domain of the corrugated disc expands as it advances, resulting in an increase in the backfill amount. Conversely, when the corrugation width remains constant, the depth of the back soil decreases as the number of corrugations increases. With an increase in the number of corrugations, the corrugated disc can more effectively eject soil during the operation, resulting in a reduction in the back soil depth.

As shown in Figure 13d, when the forward velocity remains constant, the working resistance, $y_3$, increases gradually with the number of corrugations. The reason for this is that as the number of corrugations increases, the disturbance domain of the corrugated disc grows in the forward process, resulting in an increase in the forward resistance; when the number of corrugations remains constant, the forward resistance decreases initially and then increases as the forward speed increases. With an increase in forward speed, the corrugated disc is able to dislodge the soil in the furrow during the furrow operation, and the resistance decreases marginally after increasing the soil disturbance coefficient

appropriately. When the forward speed is excessively high, the quantity of soil flung out by the corrugated disc during the forward process increases, resulting in an increase in forward resistance.

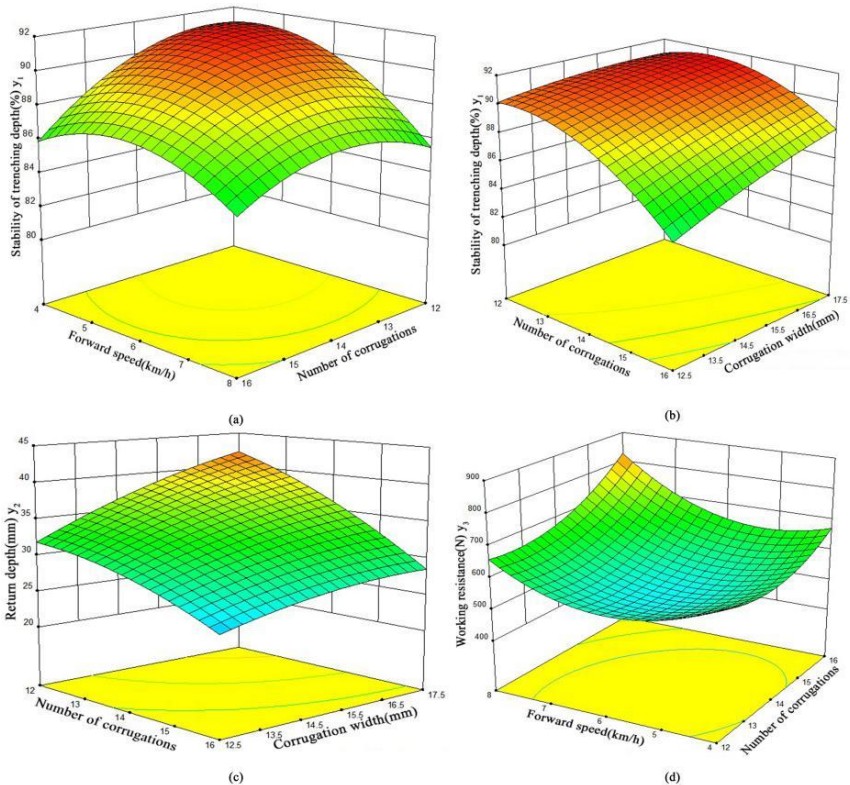

**Figure 13.** Effect of factors on furrow opener performance. (**a**) Response surface of factors A and B to $y_1$. (**b**) Response surface of factors A and C to $y_1$. (**c**) Response surface of factors A and C to $y_2$. (**d**) Response surface of factors A and B to $y_3$.

### 3.2. Parameter Optimization

The optimization module of the Design-Expert 8.0.6 software platform was applied to solve the regression model optimally with the following objectives:

$$\begin{cases} \max y_1(x_1, x_2, x_3) \\ \min y_2(x_1, x_2, x_3) \\ \min y_3(x_1, x_2, x_3) \\ s.t. \begin{cases} 12 \le x_1 \le 16 \\ 4\text{km/h} \le x_2 \le 8\text{km/h} \\ 12.5\text{mm} \le x_3 \le 17.5\text{mm} \end{cases} \end{cases} \tag{16}$$

After solving, the best overall simulation performance of the corrugated double-disc furrow opener was obtained when the furrow opener's number of corrugations was 16, the forward speed was 6 km/h, and the corrugation width was 17.5 mm. The stability coefficient of the corrugated double-disc furrow opener was 88.403% for the depth of furrowing, the depth of soil backfill was 26.19 mm, and the working resistance was 661.62 N.

To check the reliability of the results, the other parameters were kept constant, and the above optimal solution was selected for five repeatability tests. The average furrowing depth stability of the corrugated double-disc furrow opener was 88.67% for the furrowing depth, the backfill depth was 25.98 mm, and the working resistance was 662.81 N. The verification results are basically consistent with the optimization results, i.e., the optimized parameters met the design requirements [24].

### 3.3. Soil-Bin Validation Experiment Results and Analysis

The test results for the corrugated double-disc furrow opener (CDDFO) and the traditional double-disc furrow opener (TDDFO) were recorded and analyzed. The stability of the furrow depth, the area of soil disturbance, the working resistance, and the depth of soil backfill at various operating depths for the two types of furrow openers are shown in Figure 14.

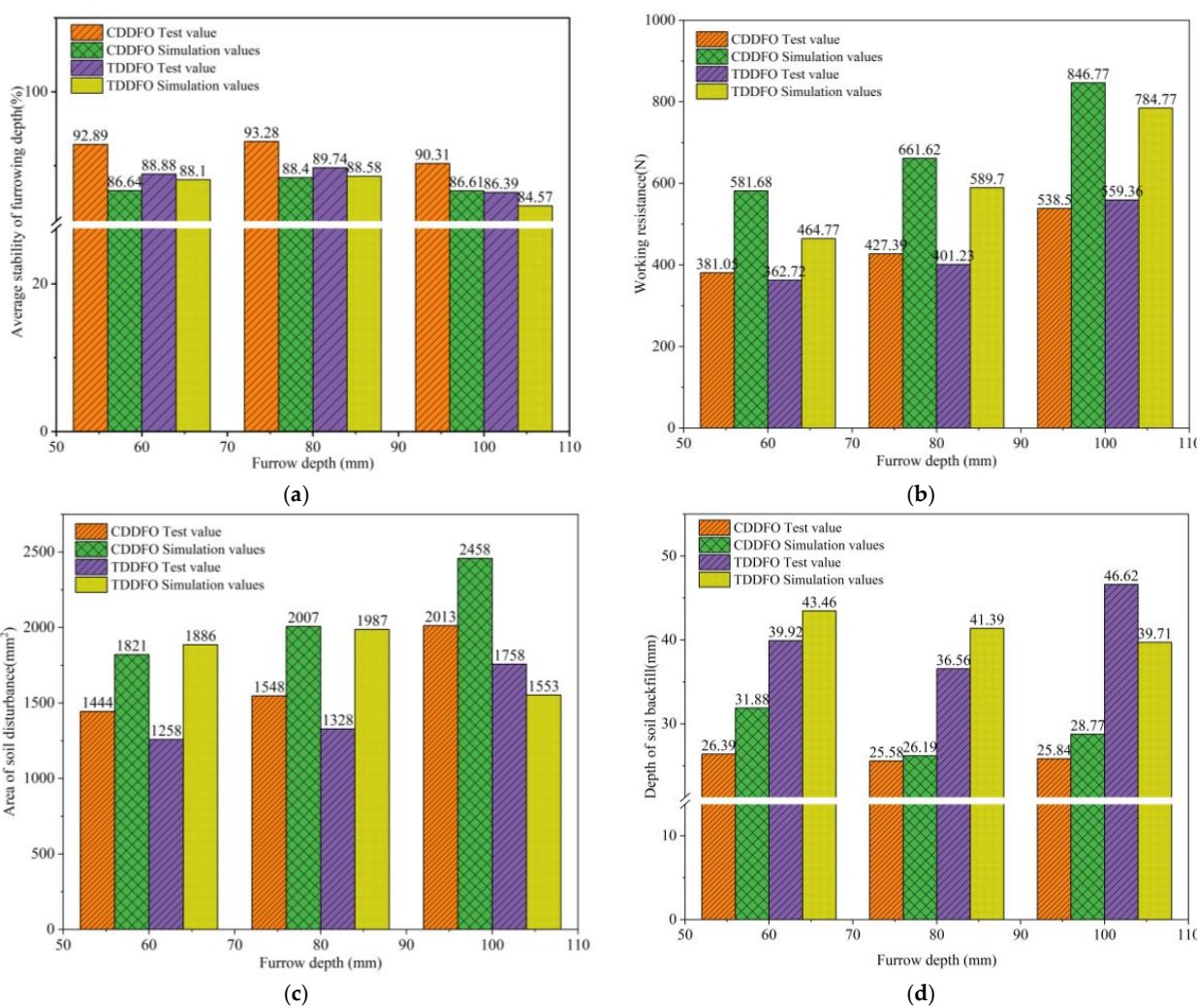

**Figure 14.** Comparison of soil bin test data. (**a**) Average stability of furrowing depth, (**b**) working resistance, (**c**) area of soil disturbance, (**d**) depth of soil backfill.

The average furrowing depth stabilities for the CDDFO and the TDDFO under a soil furrow test with a 60 mm furrowing depth operation were 92.89% (CDDFO) and 88.88% (TDDFO), respectively, which is an increase of 4.01%; the working resistances were 381.05N (CDDFO) and 362.72N (TDDFO), which is an increase of 5.1%; the average soil disturbance areas were 1444 mm$^2$ (CDDFO) and 1258 mm$^2$ (TDDFO), which is an increase of 14.79%; and the depths of soil backfill were 26.39 mm (CDDFO) and 39.92 mm (TDDFO), which is a 33.90% decrease.

As shown in Figure 14, the average furrowing depth stabilities under an 80 mm furrowing depth operation were 93.28% (CDDFO) and 89.74% (TDDFO), which is an increase of 3.54%; the working resistances were 427.39N (CDDFO) and 401.23N (TDDFO), which is an increase of 0.27%; the average soil disturbance areas were 1548 mm$^2$ (CDDFO) and 1328 mm$^2$ (TDDFO), which is an increase of 16.57%; and the soil backfill depths were 25.58 mm (CDDFO) and 36.56 mm (TDDFO), which is a decrease of 30.03%.

The average furrowing depth stabilities under a 100 mm furrowing depth operation were 90.31% (CDDFO) and 86.39% (TDDFO), which is an increase of 3.92%; the working resistances were 538.50N (CDDFO) and 559.36N (TDDFO), which is an increase of 3.73%; the average soil disturbance areas were 2013 mm$^2$ (CDDFO) and 1758 mm$^2$ (TDDFO), which is an increase of 14.51%; and the backfill depths were 25.84 mm (CDDFO) and 46.62 mm (TDDFO), which is a decrease of 44.57%.

Both of these types of furrow openers are stable and unaffected by large variations in the geometry of the furrow, making them suitable for opening seed furrows in no-till sowing. In a comparison of the operational effects of the two types of openers, the CDDFO optimizes the number of corrugated discs and the width of the corrugations based on the soil movement laws and simulation, resulting in a uniform furrow opening depth, low working resistance, and a shallow soil backfill depth. However, the extent of soil disturbance is somewhat larger. The TDDFO has poor stability of the furrowing depth, high soil disturbance, and high working resistance, and cannot work the soil out of the seed furrow, resulting in deep backfill. Combining the average values of the two varieties of furrow openers reveals that the CDDFO can enhance the uniformity of sowing depth, reduce working resistance, reduce soil disturbance, and meet the requirements of no-till sowing in orchards under green manure sowing conditions.

## 4. Discussion

With the rise in bionics and simulation, there has been extensive research on the optimal design of furrow components for field no-till seeders, and the performance of furrow openers has been significantly enhanced. Using bionics, anti-adhesion, and drag reduction mechanisms, Sun Jiyu et al. [25] created a bionic disc fissure aperture that solved the problem of adhesion in black soil in Northeast China. Jia Honglei et al. [26] introduced a chute furrow opener based on the bionics principle according to the penetrating structure optimization of a badger canine tooth surface with high efficiency and low resistance to solve the problem of the excessive resistance of a furrow opener under high-speed operation and compared it with a conventional furrow opener to reduce working resistance. However, it is challenging to adapt the furrowing form, soil disturbance, and backfill depth of the above double discs to the orchard environment in southern Xinjiang. Due to the characteristics of the low adhesion of sandy loam in southern Xinjiang and the large amount of backfill in furrowing, the existing green manure no-tillage double-disc system has some issues, including high variability in quality, poor stability of sowing depth, and bulge at the bottom of the seed furrow. According to the agricultural requirements of planting green manure in orchards [7,21], in order to meet the agricultural needs of orchards planted with green manure, a corrugated double-disc furrow opener was created by integrating the characteristics of corrugated stubble breakers and conventional double-disc furrow openers:

(1) A corrugated disc has a diameter of 250 mm, which is comparable to Ahmad Fiaz et al.'s [7] investigation. The smaller the diameter of the disc opener, the lower the working resistance, which satisfies the agronomic requirements for sowing green manure.

(2) The simulation output of soil particle trajectories indicates that the source of force for soil particles under the action of the furrow opener is primarily due to the extrusion of the furrow opener and surrounding soil. Under the action of the corrugated double-disc furrow opener, the majority of soil particles are inclined above the furrow opener in the direction of the resultant force and movement, which increases the rate of soil disturbance but effectively separates the soil in the furrow to ensure the furrow type. The direction of the resultant force and the movement speed of soil particles in the upper half of the upper furrow is inclined behind, whereas the direction of the resultant force and movement speed of soil particles in the lower half of the upper furrow is downward along the furrow wall, resulting in backfill. The result is comparable to Sun Jiyu et al.'s [25] study. Zhao Shuhong et al. [27] stated that the

backfill quantity was adequate for seed implantation. Sun Jiyu et al. [25] demonstrated that the quantity of backfill after the double-disc furrow opener destroyed the putative planting depth and the furrow depth of seeds was not conducive to seed germination. Additionally, Zhao Shuhong et al. [27] explained that the quantity of backfill was adequate for seed implantation.

(3) Through simulation testing, the structure of the double disc can be optimized further, and it can be determined that the diameter and quantity of corrugations are the most influential factors in enhancing the performance of furrowing. The results of the soil furrow experiment indicate that the working resistance of the double disc is relatively low and that the design effectively reduces the backfill depth and increases the furrowing stability coefficient while maintaining a comparable working resistance.

## 5. Conclusions

In this study, discrete element numerical simulations were used to analyze the main factors affecting the operational performance of a corrugated double-disc opener in the green manure planting segment of orchards. The corrugated disc structure was then optimized, and the design's rationality was validated through physical tests in the soil trough. The following can be concluded:

(1) Based on the agronomic requirements of planting green manure in orchard rows, a corrugated double-disc opener was designed to address the problems of a traditional double-disc opener for green manure in orchards, such as the tendency of the bottom of the furrow to bulge and the poor stability of the depth of the furrow, as well as the theoretical analysis and calculation of the structural parameters, such as the diameter of the disc, the position of the gathering point, and the position of the gathering point.

(2) This study used a discrete element numerical simulation to analyze the operation process of the furrow opener, with the goal of improving the stability of the furrowing depth, reducing the depth of soil backfill, and determining the corrugation width and the number of corrugations for the corrugated double-disc furrow opener as the primary factors by analyzing the soil movement law during furrowing operation. After optimization, the corrugated double-disc furrow opener achieved the greatest overall simulation performance when the number of corrugations was 16, the forward speed was 6 km/h, and the corrugation width was 17.5 mm.

(3) According to the agronomic requirements of green manure planting in orchards, the average furrowing stability increased by 3.54%, the working resistance and the average soil disturbance area increased by 26.16 N and 220 mm$^2$, respectively, the soil backfill depth decreased by 10.98 mm, and the straightness of furrowing significantly improved under the operating conditions of an 80 mm furrowing depth. Through the corrugation of the discs, the corrugated double-disc furrow opener devised in this paper can effectively remove soil from the seed furrow, thereby reducing the backfill depth and increasing the stability factor of the furrow opener depth. As discs are not conventionally planar, the effective contact area with soil can be enhanced, thereby increasing forward resistance.

**Author Contributions:** Resources, J.L.; data curation, J.Z.; writing—original draft preparation, R.Y. and X.M.; writing—review and editing, R.Y. and X.L.; visualization, G.S.; supervision, J.L.; project administration, J.L. and L.X. All authors have read and agreed to the published version of the manuscript.

**Funding:** This research was funded by Xinjiang Construction Corps (grant number: 2021CB022), the Finance Science and Technology Project of Alar City (2021NY07), the Finance Science and Technology Project of Alar City (2022NY13), the China Agriculture Research System of MOF and MARA (CARS-22), and the President's Foundation Innovation Research Team Project of Tarim University (TDZKCX202203).

**Institutional Review Board Statement:** Not applicable.

**Data Availability Statement:** The data presented in this study are available on request from the corresponding author.

**Acknowledgments:** We would like to thank Jiean Liao for funding the project and Xueding Ma for his help.

**Conflicts of Interest:** The authors declare no conflict of interest.

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
