# Peer review of "Optimization and Design of Disc-Type Furrow Opener of No-Till Seeder for Green Manure Crops in South Xinjiang Orchards"

_agriculture, doi:10.3390/agriculture13081474_

Round 1

Reviewer 1 Report

The manuscript’s (MS) topic deals with analyzing the soil movement between corrugated double-disc and traditional double-disc furrow openers using the discrete element method (DEM). I believe the subject of MS is interesting and necessary for increasing the effectiveness of No-till furrow openers, especially used in green manure fields. Therefore, it is new scientific evidence enough to be published in a qualified international scientific journal such as Agriculture. 

MS is enjoyable, informative, and easy to understand. Contains potentially useful information for readers. All sections of MS are well organized and presented. 

Finally, to provide some feedback to the authors, I list the following minor comments and/or concerns:

Title: I suggest the title of “Optimization and Design of Disc-type Furrow Openers of No-till Seeder for Green Manure Crops”

Abstract: When I read the Abstract I think that this research includes only simulation results and there are no verification experiments. However, the MS includes verification experiments. Therefore, the verification experiment should be mentioned in the Abstract shortly.

Line 26-27: “no-till green manure orchard seeder” is not a suitable term for identifying the seeder. I suggest using “no-till seeder for planting green manure between rows of orchards” instead.

Line 36 and 36: superscript for 2

Line 198: Isn't 80 mm seeding depth too much? What seed is sown so deep?

Line 215-220: Can you add the penetration resistance to the model parameters of soil shown in Table 1? Presenting the penetration resistance of soil simulation could be helpful in getting more meaningful results in practice.

minor revision needed

Reviewer 2 Report

The article

Research on Key Technologies and Devices for Green Manure Furrowing Operation in South Xinjiang Orchards

Was reviewed giving the following results

The article writing is poor as a result of a low clarification between simulation and soil results. The methodology should address how are the simulations data obtained and how the operation of the disks obtained. Unless the article takes form no researcher could learn something from your great work.

Line 37-38 should be eliminated in the abstract.

There is an error with citation 6 in line 59. When you begin with a citation a sentence you have to add the researcher name.

This is the same case with citation 7 in line 66.

Check also citation 8 in line 72 and citation 9 in line 81.

You repeat operation performance twice in line 79.

Line 85 is not clearly written.

F within Frictional has to be in normal letter (line 113).

G1 if gravity is not a force. Check fig 2 and eq1.

Of what experimental results you talk in Table 5. It is missing a heading at the top. Also the spacing of the Table are too big.

Line 139 is not Figure 2 it is figure 3.

There are 2 figures 12 which are separated too much

The experimental method of section 3.2 should be moved to the methodology section.

Figure 15 is repeated in 2 headings and figure 14 is missing.

Lines 398-402 should be rewritten as its editing is bad and not clear.

The average furrowing in line 404 has 2 numbers. Please rewrite paragraph and add figure number considered. The same is true for the next paragraph (lines 407-410). Please rewritte.

The authors cannot start a sentence again with a bracket and a number as lines 428 and 454.

Lines between 438 and 450 most be in one paragraph.

The conclusions have to be more directed towards the results obtained and forget requirement words in most paragraphs.

In the funding section in line 502 remove erroneous spaces.

References

In all the references the year of publication comes after the journal name.

In the author name a semicolon comes after the point and before next surname.

Capital letters are in reference 16 in the article name.

There are many editing problems.

A sentence cannot begin with a number. If this is the case please mention the first author first.

Reference years are not well positioned.

It is important to clarify the results and specify objectives and methodology.

Check number of figures as they are repeated

Reviewer 3 Report

The authors present their research on equipment for furrowing operations using DEM simulations. The authors discuss the issue using available sources and point out that the laws of soil movement are not yet sufficiently understood. From the force analysis, the geometry of a corrugated double disc opener was proposed.

The adopted input parameters were used for the DEM model. The units of the quantities are missing in Table 1. I would expect at least some basic calibration of the values used here with respect to real experiments, such as Hlosta et al. (https://doi.org/10.3390/pr8020222). Overall, I believe that the real device experiments presented in Sections 3.3.1 and 3.3.2 should be listed in the methods and performed prior to DEM simulations to validate them. But I can understand why the authors proceeded in this case. The DEM simulation was used as a support to validate the function before prototype production.

It is not entirely clear what the graphs in Figure 9 are saying. The authors should discuss them more.

A thorough revision of the whole text should be made, which contains many typos (indexing, spacing, units, references).

After minor revisions, the manuscript can be published in Agriculture.

Round 2

Reviewer 2 Report

The article

Optimization and Design of Disc-type Furrow Opener of No-Till seeder for Green Manure Crops in South Xinjiang Orchards

Has the following observations.

1.    After putting the name of the researcher and et al, [7] please avoid using points before and after the brackets.

2.    I will appreciate that figure 4 has a better resolution (line 185).

3.    The letters within Figure 5 are small maybe due to the presentation I received. Otherwise they should be increased so that the reader can view them well.

4.    The paragraph fixed within lines 275-284 is not clear. The first sentence is too long (lines 276-278) and has to get some points to get it clear.

5.    Fig. 6 in the text should appear before Fig 7. Also fig 6a and figure 6b do not appear.

6.    The letters of figure 9a and 9b are too small for the readers. The same happens for figure 10a and 10b.

7.    The letters of the axis of the 4 graphs in Fig 14 have to be increased.
